# The gap in contraceptive knowledge and use between the military and non-military populations of Kinshasa, DRC, 2016–2019

Pierre Z. Akilimali[1], Henri Engale Nzuka[2], Katherine H. LaNasa[3]*, Angéle Mavinga Wumba[2], Patrick Kayembe[1†], Janna Wisniewski[4], Jane T. Bertrand[3]

1 Kinshasa School of Public Health, Université de Kinshasa, Kinshasa, The Democratic Republic of the Congo, 2 Medical Division, Congolese Armed Forces, Kinshasa, The Democratic Republic of the Congo, 3 Department of Health Policy and Management, Tulane University School of Public Health and Tropical Medicine, New Orleans, Louisiana, United States of America, 4 Department of International Health and Sustainable Development, Tulane University School of Public Health and Tropical Medicine, New Orleans, Louisiana, United States of America

† Deceased.

* kschulze@tulane.edu

**Data Availability Statement:** All data collected under PMA2020 are made publicly available from the PMA2020 database at https://www.pmadata.

# Abstract

## Introduction

The objective of this study is to assess change over time in the modern contraceptive prevalence rate (MCPR) and related variables among married women of reproductive age (15–49 years) in the military population in Kinshasa, Democratic Republic of Congo, compared to women in the non-military population, based on cross-sectional surveys in 2016 and 2019.

## Methods

Data among women living in military camps were collected as a special study of contraceptive knowledge, use, and exposure to FP messaging, for comparison to women in the non-military population from the annual PMA2020 survey. Both used a two-stage cluster sampling design to randomly select participants. This analysis is limited to women married or in union. Bivariate and multivariate analysis was used to compare the military and non-military populations.

## Results

The socio-demographic profile of women in the military camps differed between 2016 and 2019, which may reflect the more mobile nature of this population. In both populations, knowledge of modern contraceptive methods increased significantly. Similarly, use of a modern contraceptive method also increased significantly in both, though by 2019 women in the military camps were less likely to use modern contraception (24.9%) than their non-military counterparts (29.7%). Multivariate analysis showed no significant difference in the amount of increase in MCPR for the two populations. Among contraceptive users in both populations, the implant was the leading method. Potential effects of FP programming were evident in the military population: exposure to FP messaging increased (in comparison to a

org/data/available-datasets. All data for the two military surveys have been added to our submission as supplementary file 6.

**Funding:** Financial support for this research came from the Bill and Melinda Gates Foundation (INV#-009285, INV#-007330), https://www.gatesfoundation.org/. All authors (PZA, HNE, KHS, AWM, PAK, JW, JTB) received partial salary support or consulting fees from this grant for this research. The funders had no role in study design, data collection and analysis, decision to publish, or preparation of the manuscript.

**Competing interests:** The authors have declared that no competing interests exist.

decrease among the non-military population). Moreover, women who had lived in the camps for 4+ years had a higher MCPR than those living in the camps for less than four years.

## Conclusions

This study demonstrates the feasibility and importance of collecting data in military camps for better understanding contraceptive dynamics among this specialized population.

## Introduction

The Democratic Republic of the Congo has one of the highest total fertility rates in the world: 6.2 children [1], accompanied by a low level of modern contraceptive use (17.8%) among women married or in union, as of 2018 [1]. Kinshasa, the capital city, has shown the usual effects of urbanization on family size through declining fertility rates within the urban area compared to rural areas, as observed in other settings [2, 3]. As of 2018, the total fertility rate for Kinshasa was 3.6 children [1], and modern contraceptive prevalence among women married or in union had risen to 29.6% [4, 5].

Whereas contraceptive analyses often focus on a specific sub-group within the larger population (e.g., adolescents, indigenous, or other groups known to differ from the general public), little research is available from low- and middle-income countries on military populations. In 2016 we conducted a survey of women living in military camps in the city of Kinshasa [6], which indicated a much lower level of modern contraceptive use (10.9%) than among the non-military population (20.9%), studied the same year through a different survey [6]. These differences could not be explained by socio-demographic factors alone; the two populations were similar on age distribution, marital status, level of education, wealth index, and number of lifetime births.

The authors conducted a search of the PubMed database using the key terms "Family Planning," "Contraception," "Military," "Armed Forces," and "Police." The search results did not reveal any relevant articles published after the results of the 2016 survey among women living in military camps in Kinshasa, DRC were published [6]. However, there are three older studies in Nigerian military barracks which were relevant to this paper. A survey among male soldiers in Nigeria found that use of contraception was significantly lower than the national average, despite nearly 75% of men reporting they approved of using contraception [7]. Another study among adolescents living in military barracks in Nigeria found very low knowledge about contraceptive methods besides condoms [8]. An educational intervention on family planning (FP) knowledge, attitudes and practices among married women living in military barracks in Nigeria also found increases in the mean number of contraceptive methods known and use of modern contraception [9].

Since 2016, the level of FP service delivery has intensified in Kinshasa. The percentage of facilities that offer FP has steadily increased [5]. Additionally, several organizations have established or increased community-based distribution programming; nursing students now provide contraceptives through community outreach activities as part of their nursing school training [10, 11]; initiatives that target adolescents and post-partum women have been launched. Thus, it is not surprising that the modern contraceptive prevalence rate (MCPR) in Kinshasa has increased in a stepwise fashion in recent years [5].

The analysis in this article assesses changes over time in contraceptive dynamics among the women in the military population in Kinshasa, based on two cross-sectional surveys in 2016

and 2019. The PMA2020 surveys (renamed PMA as of 2019) are conducted annually among the general population in Kinshasa, therefore, we can compare the changes found in the military population to those found in the PMA2020 surveys among the non-military population. The specific objectives of this analysis are:

1. To assess the extent of change in modern contraceptive use between 2016 and 2019 in the military camps, and to compare it to change in the non-military population;

2. To assess the change in contraceptive method mix between 2016 and 2019 in the military camps, and compare it to the non-military population;

3. To measure exposure of the military and non-military population to FP programming as of 2019; and

4. To identify the factors associated with modern contraceptive use in both the military and the non-military population as of 2019.

## Methodology

### Setting

The city of Kinshasa consists of 35 health zones, including three which are highly rural and distant from the center of the city (thus, atypical). The military health zone of Kokolo consists of 15 non-adjacent "camps," which operate as isolated settlements within the boundaries of the highly populated areas of the city. Fourteen of the 15 camps have a health facility, operated by the military for the residents of the camp.

Prior to the 2016 survey, some of the health facilities within Kokolo sporadically had contraceptives available, but stockouts were frequent and the range of methods was limited. No group education sessions or other types of community awareness raising took place. However, as of 2016, the military established a Coordination Unit for Reproductive Health for the military camps. Numerous activities were conducted to reinforce the provision of FP services: training of clinical personnel within the military-run health facilities, training of community-health workers to serve as community-based distributors (CBD), regular resupply of facilities and CBD with a full range of contraceptives, provision of materials needed for the administration of implants and IUDs, mini-campaigns (outreach activities) several times a year that heightened the visibility and improved access to contraceptives, and promotional materials (such as the FP billboard outside the gates of the camp depicting a military family, shown in Fig 1).

In the non-military health zones of Kinshasa, FP service provision varies by health zone, often depending on support from international NGOs; there is no "standard treatment" citywide. The non-military health zones generally benefit from the same combination of activities as the military health zones, with a less systematic or regular delivery. In short, the programmatic intervention in the military zone starting in 2016 was designed to ensure the same service delivery inputs as many of the non-military health zones were already receiving, albeit in more piecemeal fashion.

### Data

This analysis uses data collected in 2016 and 2019–2020 from two sources. The first source of data is the Performance Monitoring and Accountability Project (PMA) which can be used to represent the non-military population in Kinshasa. PMA measures factors related to FP services, FP messaging and communication, contraceptive uptake and use, and contraceptive

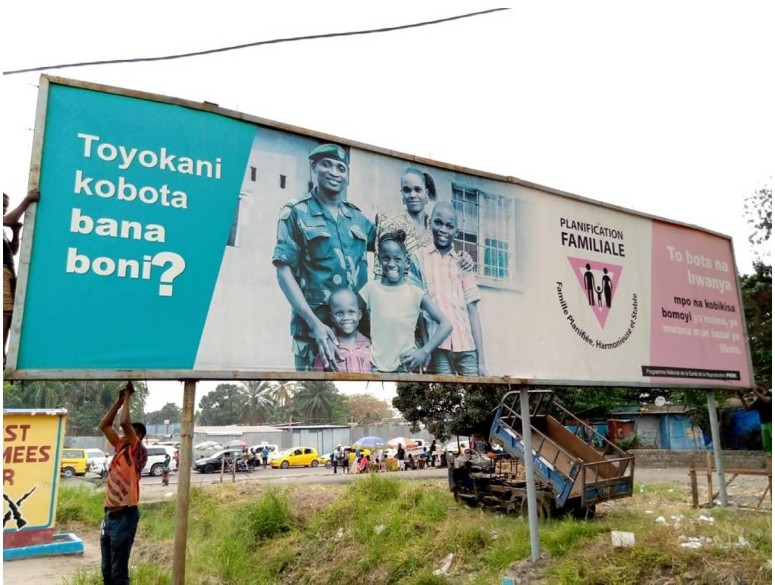

**Fig 1. Family planning billboard depicting military personnel, located outside the military camp in Kinshasa, DRC.**

decision-making in 11 countries across Africa and Asia. The survey uses a two-stage cluster sampling approach to achieve a representative sample of the province. First, census enumeration areas (EAs) are randomly selected, using selection probabilities proportional to EA size, within the province. Next, all households within the selected EAs are listed and 33 households are randomly selected for the survey. At each selected house, the head of the household is first asked to complete a household survey, then all resident women of reproductive age (15–49 years) are interviewed. The PMA female survey, which is conducted with only female interviewers, consists of basic demographic information and in-depth information on fertility history and preferences and use of contraception. In the DRC, eight rounds of data collection have occurred in Kinshasa (2013–2020). For this analysis, data from 2016 (round five) and 2019–2020 (round eight) are used; different units of residence were selected in the two surveys. In the 2016 survey round, 2,582 women were interviewed between September and October 2016. In the 2019–2020 survey round, 2,611 women were interviewed between December 2019 and February 2020. (Because the interval between surveys was closer to three than four years, we have labeled the round 8 survey as occurring in 2019 for simplification.) Our analysis is restricted to only married women, so in total, 1,288 women from the 2016 PMA survey and 1,085 women from the 2019–2020 PMA survey are included in this study.

The second data source is a separate survey among the military population, conducted at an interval of three years: from Nov 19-Dec 12, 2016, and from Dec 29, 2019-Jan 21, 2020. (Again, for simplicity, we have labeled the latter as the "2019 survey.") To capture a representative sample of this population, a similar two-stage cluster design was used. Ten military camps were randomly selected, using selection probability based on population size, out of the 15 camps in Kinshasa. The selected camps were then divided in EAs, and one EA was randomly selected in each of the 10 camps. Following the PMA format, the households in each selected EA were listed and 33 household were randomly selected. The same units of residence were used in both military surveys, although the residents could have changed. In the 2016 military survey, all women of reproductive age (15–49 years) living in a selected household were interviewed; in the 2019 survey only women 15–49 married or in union were interviewed, since we

planned to limit this analysis to women married or living in union. The cases available for analysis were n = 229 in 2016, n = 357 in 2019. Based on the sampling techniques used, the data can be considered generalizable to women in the military populations and the non-military populations in Kinshasa.

The content and procedures for the PMA and military surveys were highly similar on the following points. The local study directors were the same for both surveys. The large majority of the questions in the military surveys were taken verbatim from the PMA questionnaire, with the same terminology and response options, except for a few additional questions not found in the PMA questionnaire. The questions were administered by trained female interviewers in French or Lingala (the local language). Data were collected in both rounds and both populations using tablets or smartphones programmed with ODK.

All women provided written and informed consent prior to taking part in this study. Parental consent is not required to interview women ages 15–17 in the DRC as they are considered adults in this setting. The data collection among the military population was approved by the Tulane IRB (study 492318) and the Kinshasa SPH (#ESP/CE/070/2017). The PMA2020 data were publicly available from the PMA website, having undergone IRB review at the Johns Hopkins Bloomberg School of Public Health (#14702).

## Measures

A number of socio-demographic characteristics were examined including age, level of education, and husband having other wives. Household wealth was only included in the PMA surveys and the 2016 military survey. This measure was constructed using a wealth index based on ownership of 25 household assets, house material, livestock ownership and water source, which was converted into quintiles. Whether a woman's husband was living with her was only assessed among the military population.

A series of measures focused on women's reproductive history were included in our analysis. These measures consisted of lifetime number of births, mean age at first sex, last birth intended, and desire for another child. Experience of child death was only asked of the military population and the 2016 PMA survey.

FP use was measured as use of any method to prevent pregnancy at the time of the survey, then divided into use of modern and traditional methods. Modern methods included male and female condoms, pills, injectables, implants, IUD, female sterilization, emergency contraception, CycleBeads, and lactational amenorrhea method (LAM). Traditional methods included rhythm, withdrawal, and other traditional methods (e.g., folkloric methods such as amulets, herbs). Use of modern contraception was further divided into use of long acting reversible (LARC) methods, which include implants and IUD. Type of method currently used was assessed to measure method mix.

Finally, the survey measured awareness of FP and exposure to FP messages. Knowledge of contraceptive methods was measured by whether a woman had heard of condoms, pills, injectables, implants, IUD, CycleBeads, emergency contraception, rhythm method and withdrawal. Five indicators were used to measure exposure to FP in both populations: whether the woman had been visited by a FP health worker in the past 12 months, if a staff member at a health facility has talked with her about FP methods in the past 12 months, and whether the woman had seen something about FP on the TV, heard about FP on the radio, or read something about FP in a magazine or newspaper. These five measures were then combined to create a score for the number of sources of FP message exposure. In addition, two questions were asked of military women in both years and non-military women only in 2016: whether they had seen the FP logo and if they had participated in a group talk about FP at the community level. Finally, only

military women were asked about their exposure to the FP billboard depicting a military family.

## Analysis

Weighted frequencies for all explanatory variables were calculated. Data were analyzed using STATA 14.1 (Stata Corp, College Station, Texas, US). First, the background sociodemographic characteristics and reproductive history of military and non-military women in both years were tabulated using weights to account for the sampling design and clustering within EAs. Mean age and mean age at first sex were compared using two-tailed t-test and all other variables were compared using chi-squared test. Second, bivariate analysis using the chi-squared test was conducted to compare FP knowledge and exposure measures in 2016 and 2019 among military and non-military women. Next, contraceptive use and method mix were compared between years among military and non-military women using chi-squared bivariate analysis. Logistic regression generalized linear latent and mixed models (GLLAMM) with the logit link and binomial family [12] that adjusted for clustering and sampling weights was used to measure the level of association between use of modern contraception and explanatory variables (controlling for sociodemographic characteristics, reproductive history, and exposure to FP programming and messaging). We introduced the Interaction term between population and year in the model, to explore the difference in the change in modern contraceptive use among military women compared to non-military women between 2016 and 2019. Chi-squared test was used to compare the use of modern contraception among women living in military camps in 2019 based on the duration of time in the camp, four or more years and less than four years, compared to women in the military population in 2016. All results were assessed using an alpha level of 0.05.

## Results

### Socio-demographic profile

Given that differences in socio-demographic characteristics of the samples in 2016 versus 2019 can potentially influence levels of contraceptive use, we begin by comparing the samples of the two populations between the two surveys in Table 1. All results are based on women married or in union aged 15–49 years old.

Women in the military population differed across the two surveys on several characteristics. In comparison to the 2016 sample, the 2019 sample was younger by almost 2 years (31.3 years), had fewer lifetime births (2.8), and had initiated sex at a slightly earlier age (16.2 years). There were no significant differences in level of education (with 66.5% reporting some level of completion of secondary school) or the husband having other wives (10.6%). However, in contrast to 2016 when most women (93.1%) reported that their husbands were living with them, this percent dropped to 57.0% in 2019 (Table 1).

Of the women interviewed in the military camps, only 5.0% (in 2016) and 3.5% (2019) were enlisted in the military themselves, the remaining women interviewed were wives, daughters or other family members of military personnel. As of 2019, four in five of the women in the military population (79.6%) had lived in the camps for 4+ years, compared to 20.4% who had lived there less than 4 years. The vast majority (94.8%) of women living in the military camps had lived in Kinshasa for 4+ years. Forty percent (40.1%) reported having been interviewed in 2016.

Among the non-military population, we found small but significant changes between the two samples, although on different variables. In 2019, the percentage with some level of secondary school had increased to 72.5%; fewer women (7.7%) reported that their husbands had

**Table 1. Weighted background characteristics for military and non-military women, married or in union, Kinshasa 2016 and 2019.**

| | Military[1] | | | | Non-Military | | | |
|---|---|---|---|---|---|---|---|---|
| | **2016** | **2019** | **Diff.** | **p-value** | **2016** | **2019** | **Diff.** | **p-value** |
| | **n = 229** | **n = 357** | | | **n = 1288** | **n = 1085** | | |
| Socio-demographic characteristics | | | | | | | | |
| Age | | | | <**0.001** | | | | 0.123 |
| 15–19 | 0.4 | 10.7 | 10.3 | | 2.8 | 2.1 | -0.7 | |
| 20–24 | 6.4 | 16.9 | 10.5 | | 11.6 | 10.7 | -0.9 | |
| 25–29 | 24.3 | 16.7 | -7.6 | | 19.0 | 20.4 | 1.4 | |
| 30–34 | 27.8 | 19.5 | -8.3 | | 21.2 | 19.7 | -1.5 | |
| 35–39 | 22.7 | 16.7 | -6.0 | | 20.5 | 17.8 | -2.7 | |
| 44–44 | 11.0 | 10.8 | -0.2 | | 15.0 | 18.4 | 3.4 | |
| 45–49 | 7.3 | 8.7 | 1.4 | | 10.0 | 10.8 | 0.8 | |
| Mean age (std. dev.) | 33.2 (6.9) | 31.3 (8.8) | 1.9 | **0.003** | 33.5 (7.8) | 33.9 (7.9) | 0.4 | 0.141 |
| Level of education | | | | 0.389 | | | | <**0.001** |
| None/Primary | 19.4 | 15.6 | -3.8 | | 23.7 | 11.0 | -12.7 | |
| Some level of secondary | 65.8 | 66.5 | 0.7 | | 60.6 | 72.5 | 11.9 | |
| Higher than secondary | 14.8 | 17.9 | 3.1 | | 15.7 | 16.4 | 0.7 | |
| % of female respondents who are themselves in the military | 5.0 | 3.5 | 1.5 | 0.378 | -- | -- | -- | |
| Wealth index[2] | | | | | | | | 0.269 |
| Quintile 1 | 25.8 | -- | -- | | 19.6 | 17.8 | -1.8 | |
| Quintile 2 | 28.3 | -- | -- | | 19.0 | 17.2 | -1.8 | |
| Quintile 3 | 22.9 | -- | -- | | 20.1 | 19.4 | -0.7 | |
| Quintile 4 | 14.2 | -- | -- | | 20.7 | 22.3 | 1.6 | |
| Quintile 5 | 8.9 | -- | -- | | 20.6 | 23.4 | 2.8 | |
| Husband has other wives | 12.4 | 10.6 | -1.8 | 0.499 | 10.0 | 7.7 | -2.3 | **0.054** |
| Husband living with her[2] | 93.1 | 57.0 | -36.1 | <**0.001** | 87.5 | -- | -- | |
| Reproductive history | | | | | | | | |
| Mean Age at first sex (std. dev.) | 17.0(2.5) | 16.2(2.9) | 0.7 | **0.007** | 17.0(2.9) | 17.5(3.3) | 0.5 | <**0.001** |
| Number of lifetime births | 3.3 | 2.8 | -0.5 | **0.008** | 3.16 | 3.33 | 0.17 | **0.042** |
| Experienced a child death[2] | 10.6 | 25.3 | 14.7 | <**0.001** | 21.2 | -- | -- | |
| Last birth intended | 38.8 | 45.5 | 6.7 | 0.159 | 58.1 | 55.2 | -2.9 | 0.197 |
| Does not want another child | 45.5 | 31.6 | -13.9 | **0.001** | 30.1 | 27.5 | -2.6 | 0.167 |
| Additional data on women in military camps—2019 only (n = 357) | | | | | | | | |
| Interviewed in 2016 | -- | 40.1 | -- | | -- | -- | -- | |
| Birthplace: | | | | | | | | |
| Kinshasa | -- | 59.9 | -- | | -- | -- | -- | |
| Elsewhere | -- | 40.1 | -- | | -- | -- | -- | |
| Residence in mil. camps | | | | | | | | |
| < 4 years | -- | 20.4 | -- | | -- | -- | -- | |
| 4+ years | -- | 79.6 | -- | | -- | -- | -- | |
| Residence in Kinshasa | | | | | | | | |
| < 4 years | -- | 5.2 | -- | | -- | -- | -- | |
| 4+ years | -- | 94.8 | -- | | -- | -- | -- | |

[1]All values are presented as percentages except for age, which is shown as the mean (with its standard deviation);

[2]variable was not collected in 2019 among military population;

[3] variable was not collected in this round of PMA 2019.

other wives; they had slightly higher number of live births (3.3); and the age at first sex had increased to 17.5 years. Non-military women in the 2019 sample did not differ significantly from those in the 2016 sample on age (33.5 versus 33.9 years).

## Knowledge of contraceptive methods

In populations with relatively low contraceptive use, knowledge of contraceptive methods is a useful indicator to gauge progress regarding the introduction of FP. As shown in Table 2, the mean number of contraceptive methods known increased significantly in both populations. However, overall, military women in 2019 were still less knowledgeable about contraceptive methods than their non-military counterparts; military women knew a mean number of 8.5 methods (out of ten) versus non-military women knew all ten methods (p = 0.026).

Among the military women, increases in knowledge were significant for implants, inject-ables, and CycleBeads. By contrast, fewer military women reported knowledge of female con-doms, and the percentages reporting knowledge of the two traditional methods (rhythm and withdrawal) were strikingly lower in the 2019 sample. Among the non-military women, knowledge significantly increased for implants, pills, emergency contraception (by far the larg-est increase), male condoms, CycleBeads, and rhythm. The only exception was a slight

**Table 2. Knowledge of contraceptive methods and exposure to FP programming among women married or in union, by type of population and year of survey.**

|  | Military | | | | Non-Military | | | |
|---|---|---|---|---|---|---|---|---|
|  | 2016 | 2019 | Diff. | p-value | 2016 | 2019 | Diff. | p-value |
|  | n = 229 | n = 357 | | | n = 1288 | n = 1085 | | |
| Knowledge: % has heard of: | | | | | | | | |
| Male condoms | 85.6 | 87.5 | 1.9 | 0.475 | 97.4 | 98.6 | 1.2 | **0.031** |
| Female condoms | 82.5 | 74.3 | -8.2 | **0.026** | 82.9 | 79.6 | -3.3 | **0.038** |
| Pills | 80.8 | 84.0 | 3.2 | 0.345 | 87.3 | 90.1 | 2.8 | **0.035** |
| Injectables | 72.2 | 81.3 | 9.1 | **0.012** | 94.4 | 95.5 | 1.1 | 0.178 |
| Implants | 70.1 | 89.0 | 18.9 | **<0.001** | 92.4 | 96.8 | 4.4 | **<0.001** |
| IUD | 58.8 | 66.8 | 8.0 | 0.056 | 61.7 | 64.8 | 3.1 | 0.114 |
| CycleBeads | 56.3 | 72.6 | 16.3 | **<0.001** | 58.0 | 70.5 | 12.5 | **<0.001** |
| Rhythm method | 54.2 | 29.3 | -24.9 | **<0.001** | 94.8 | 97.9 | 3.1 | **<0.001** |
| Emergency contraceptive | 39.1 | 37.7 | -1.4 | 0.731 | 33.7 | 60.0 | 26.3 | **<0.001** |
| Withdrawal | 58.3 | 52.8 | -5.5 | 0.195 | 95.0 | 96.2 | 1.2 | 0.175 |
| Mean # of methods known | 8.0 | 8.5 | 0.5 | **<0.001** | 9.5 | 10.0 | 0.5 | **<0.001** |
| Exposure to FP (%): | | | | | | | | |
| Saw something about FP on the TV | 69.6 | 85.8 | 16.2 | **<0.001** | 64.4 | 55.4 | -9 | **<0.001** |
| Heard about FP on radio | 67.6 | 80.2 | 12.6 | **0.001** | 39.6 | 35.1 | -4.5 | **0.022** |
| Staff member at health facility speaks to you about FP methods in the past 12 months | 20.3 | 24.9 | 4.3 | 0.208 | 21.6 | 18.2 | -3.4 | **0.028** |
| Read about FP in a magazine/newspaper | 58.1 | 45.3 | -12.8 | **0.003** | 13.1 | 11.6 | -1.5 | 0.259 |
| Visited by Health worker in the past 12 months | 29.5 | 40.8 | 11.3 | **0.007** | 8.8 | 4.8 | -4 | **<0.001** |
| Has seen the FP logo[1] | 88.0 | 97.9 | 9.9 | **<0.001** | 49.6 | – | – | |
| Saw FP billboard showing military family[2] | 91.5 | 97.9 | 6.4 | **<0.001** | – | – | – | |
| Participated in a group talk at the community level about FP[3] | 12.2 | 27.6 | 15.4 | **<0.001** | 17.7 | – | – | |
| Score: number of sources of FP message exposure[4] | 2.6 | 2.8 | 0.2 | **0.023** | 1.6 | 1.4 | -0.2 | **<0.001** |

(1) Was not collected in the last round of PMA2020;

(2): was collected only in military population;

(3): was collected only in military population in 2019;

(4) Based on the five variables that were collected in both populations on both surveys, listed in this table.

decrease in knowledge of female condoms. The lower knowledge of female condoms than other methods reflects more limited availability of this method in many service delivery sites.

## Exposure to family planning messaging and personnel

In 2016 and 2019, we asked respondents in both populations about exposure to five types of FP messaging or interaction with health workers. Three additional questions were asked on both rounds to the military only. The trends are striking (Table 2). Among the military populations, the percentage of women exposed to different messages or visits between the two surveys increased significantly on four of the five items. By contrast, in the non-military population the percentages of exposure significantly decreased by 3–9 percentage points on four of the five items. On all three additional items asked only of the military population on both rounds, increases were again significant. By 2019, 97.9% of military women had seen the FP logo and the billboard that showed a military family (see Fig 1). The percentage that had participated in a group talk at the community level about FP more than doubled from 12.2% to 27.6%.

## Modern contraceptive use and method mix

The key outcome of interest in this analysis is modern contraceptive use. As shown in Table 3, use of modern contraception increased in both populations between 2016 and 2019. The percentage of married women in military camps using a modern contraceptive method was significantly higher in 2019 (24.2%) than in 2016 (15.0%). Among these women, use of a long-acting

**Table 3. Contraceptive use and method mix among women married or in union in Kinshasa, by type of population and year of survey.**

| | Military | | | | Non-Military | | | |
|---|---|---|---|---|---|---|---|---|
| | **2016** | **2019** | **Diff.** | **p-value** | **2016** | **2019** | **Diff.** | **p-value** |
| | **n = 228** | **n = 358** | | | **n = 1252** | **n = 1086** | | |
| Current use of any contraceptive method (%) | 23.0 | 32.2 | 9.2 | **0.021** | 49.8 | 54.2 | 4.4 | **0.033** |
| Current use of modern contraceptive method (%) | 15.0 | 24.2 | 9.2 | **0.007** | 23.1 | 29.7 | 6.6 | **<0.001** |
| Current use of long-acting contraceptive method (%) | 7.0 | 12.2 | 5.2 | 0.060 | 8.0 | 13.3 | 5.3 | **<0.001** |
| Current use of traditional contraceptive method (%) | 8.0 | 8.0 | 0.0 | 0.982 | 26.7 | 24.5 | -2.2 | 0.200 |
| Obtained desired method (%) | 66.9 | 79.7 | 12.8 | **0.025** | 92.3 | 97.1 | 4.8 | **<0.001** |
| | Among women using any contraceptive method | | | | | | | |
| | n = 54 | n = 120 | | | n = 624 | n = 571 | | |
| Current method used (%) | | | | | | | | |
| Modern methods: | | | | | | | | |
| Implant | 21.2 | 35.8 | 14.6 | | 13.7 | 22.7 | 9.0 | |
| Male condoms | 18.0 | 4.8 | -13.2 | | 12.9 | 10.0 | -2.9 | |
| Injectable | 13.8 | 14.7 | 0.9 | | 6.9 | 5.5 | -1.4 | |
| Female sterilization | 5.8 | 0.7 | -5.1 | | 0.9 | 2.0 | 1.1 | |
| IUD | 4.0 | 1.5 | -2.5 | | 2.0 | 0.7 | -1.3 | |
| CycleBeads | 2.2 | 3.8 | 1.6 | | 0.8 | 1.3 | 0.5 | |
| Pills | 0.0 | 3.9 | 3.9 | | 7.0 | 3.2 | -3.8 | |
| Female condoms | 0.0 | 4.8 | 4.8 | | 0.0 | 0.2 | 0.2 | |
| Emergency contraception | 0.0 | 2.8 | 2.8 | | 1.7 | 7.9 | 6.2 | |
| LAM | 0.0 | 2.3 | 2.1 | | 0.4 | 0.0 | -0.4 | |
| Traditional methods: | | | | | | | | |
| Rhythm | 16.6 | 6.9 | -9.7 | | 33.6 | 33.0 | -0.7 | |
| Withdrawal | 9.0 | 12.3 | 3.3 | | 14.9 | 10.7 | -4.1 | |
| Other traditional | 9.4 | 5.7 | -4.2 | | 5.2 | 2.8 | -2.3 | |

reversible contraceptive (LARC) increased from 7.0% to 12.2% while use of a traditional method remained constant at 8.0%. The percentage of contraceptive users living in military areas who were able to obtain their desired contraceptive method significantly increased by 12.8 percentage points to 79.7%.

A similar pattern is observed in the non-military population. Among these women, modern contraceptive use significantly increased from 23.1% to 29.7%, and use of LARCs also significantly increased from 8.0% to 13.3%. Although there was no significant change in use of traditional methods, the level of traditional use remained relatively high (24.5%), much higher than in the military population. The percentage of women in the non-military population who were able to obtain their desired method of contraception significantly increased by nearly 5 percentage points to 97.1%.

The method mix has also changed between 2016 and 2019 in the two populations. Among users in both the military and non-military populations, implants showed the greatest increase and represented by far the most widely used modern contraceptive. Among the military population, several methods not in use in 2016 (oral pills, female condoms, emergency contraception, and lactational amenorrhea method) were reported in small percentages (2.3–4.8%) in 2019. Also, among the military, there was a sharp drop (-13.2 percentage points) in male condom use between surveys.

Our analysis also examined two other variables related to contraceptive use. The first was whether the last birth was intended (Table 1). Although no significant differences occurred in either population between the two surveys, the non-military women in 2019 were more likely (55.2%) than those in the military population (45.5%) to report that their last birth was intended. This finding is consistent with the lower levels of knowledge and use among the women in the military population. By contrast, the percentage expressing desire for another child (Table 1) was significantly lower in 2019 (31.6%) than in 2016 (45.5%) among the military, whereas among the non-military population, it remained unchanged.

We examined whether the gap in MCPR between the two populations had narrowed significantly between the two surveys, shown in Table 4. Multivariate regression analysis was used to compare use of modern contraception between military and non-military women while controlling for sociodemographic characteristics (age, education, number of live births, husband having other wives) and exposure to FP service delivery and programming (six factors shown in Table 2 available across all data sets). Overall, women in 2019 were significantly more likely to use modern contraception than in 2016 (OR 2.2, 95% CI 1.52 to 3.06). The change in use of modern contraception between survey years did not significantly differ between military and non-military women. In short, MCPR increased in both populations, with no significant narrowing of the gap between the two. Two of four socio-demographic variables did influence modern contraceptive use: age and number of live births; by contrast, two others (level of education and husband having other wives) did not. Of three variables related to communication, the mean number of channels on which respondents had seen FP messages and having been visited by a health worker in the past 12 months influenced modern contraceptive use but having spoken to a staff member at a health facility or did not. Desire for another child was also negatively associated with modern contraceptive use.

Finally, given the differences in the socio-demographic characteristics of the military population across the two surveys, we examined the level of MCPR for women in the 2019 sample who had lived in the military camp for less than four years versus those living there for four or more years compared to the 2016 military sample. Had the FP activities been effective in increasing MCPR among the military population, we would expect higher MPCR among those women having lived in the military camps for four or more years.

**Table 4. Adjusted Odd Ratios (OR) for factors associated with modern contraceptive use in Kinshasa, 2016–2019.**

| Explanatory variables | Uses modern contraception | | |
|---|---|---|---|
| | OR | 95% CI | p-value |
| Population | | | |
| Non-Military | 1.00 | | |
| Military | 0.37 | 0.16–0.85 | **0.018** |
| Year of Survey | | | |
| 2016 | 1.00 | | |
| 2019 | 2.15 | 1.52–3.06 | **<0.001** |
| Interaction term (Population*Year) | 1.14 | 0.33–3.83 | 0.838 |
| Age | 0.92 | 0.90–0.95 | **<0.001** |
| Number of live births | 1.18 | 1.08–1.29 | **<0.001** |
| Desire for another child(ren) | | | |
| Yes | 1.00 | | |
| No | 1.88 | 1.21–2.93 | **0.005** |
| Level of education | | | |
| None/Primary | 1.00 | | |
| Middle/secondary | 0.90 | 0.63–1.28 | 0.552 |
| Tertiary | 1.43 | 0.86–2.36 | 0.165 |
| Husband has other wives | | | |
| No | 1.00 | | |
| Yes | 1.03 | 0.64–1.68 | 0.897 |
| Staff member at health facility spoke to you about FP methods in the past 12 months | | | |
| No | 1.00 | | |
| Yes | 1.40 | 0.90–2.18 | 0.131 |
| Saw FP messages via the media (score) | 1.24 | 1.01–1.51 | **0.035** |
| Visited by Health worker in the past 12 months | | | |
| No | 1.00 | | |
| Yes | 1.71 | 1.14–2.57 | **0.010** |
| Constant | 1.03 | 0.48–2.23 | 0.937 |
| Log likelihood | **-1313.20** | | |
| Number of level 1 units | **2517** | | |
| Number of level 2 units | **66** | | |

Fig 2 confirms this premise. By 2019, MCPR was 25.9% among those living in the camps for 4+ years (p = .001) compared to 20.9% among those living in the camps for less than four years (p = 0.241).

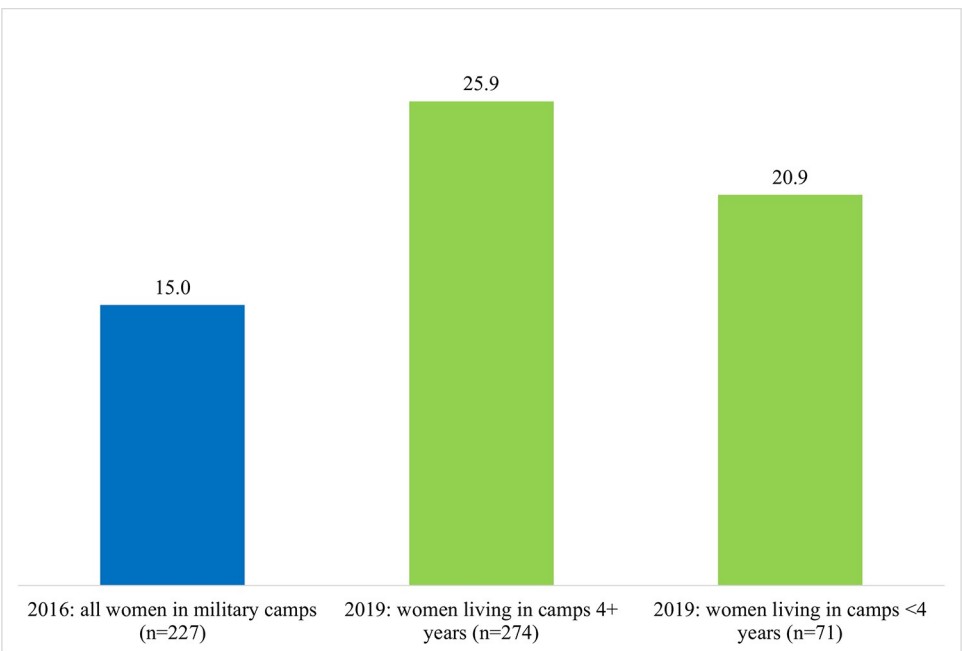

**Fig 2. Differences in modern contraceptive prevalence between surveys by length of residence in the military camps (military women only).**

## Discussion

The military population in the DRC is a sub-population of interest for FP programming for several reasons. First, military personnel live in specific geographic areas known as camps, although these camps are not always adjacent. Second, the military represents a captive audience for any health intervention, although all FP services are provided on a voluntary basis. Third, military personnel are often deployed to other military camps in other parts of the country, which may have a multiplier effect across the country. Finally, similar to studies among military families in Nigeria [7], use of contraception among women living in military camps in Kinshasa is significantly lower than the general population, indicating a potentially underserved population for FP programming.

The composition of the women respondents in the military population differed on several variables between 2016 and 2019 (e.g., fewer women living with their husbands), whereas the characteristics of the non-military population showed less change. One reason may be the higher mobility and frequent deployment of military families to different camps across the country, resulting in a population with slightly different characteristics. Also, the sample size was far larger for the non-military respondents, contributing to less variability across years.

Results show that modern contraceptive use in Kinshasa was higher in 2019 than 2016 among both the military and non-military populations. Similar to the results observed in the 2016 surveys, in 2019 married women in the non-military population (29.7%) had higher levels of contraceptive use than in the military population (24.2%).

Results from the 2019 survey indicate the method mix is diversifying among the military population. Four new contraceptive methods were reported in small percentages in 2019 which were not present in the 2016 survey. The notable decline in use of male condoms between surveys suggest uptake of more effective methods. Among the non-military population, use of long-acting reversible methods increased significantly, and there was an observed

increase among the military population as well. This trend can be explained by a dramatic increase in use of implants among both populations. The growing use of implants mirrors the increasing preference for this method at the national level in the DRC [1] as well as in other sub-Saharan countries [13]. Curiously, the percentage of women continuing to rely on traditional methods of contraceptive use was far higher among women in the non-military population (~25%) than in the military population (only 8.0%). The widespread reliance on traditional methods–at the same time as use of modern contraception continues to rise–remains an anathema to many program managers. The reason appears to lie in the deep-seated fear that modern methods may jeopardize a woman's future fertility [14]. These method mix results differ from those in the FP intervention in Nigerian military camps where no women reported using implants or traditional methods [9].

Several factors were found to be associated with modern contraceptive use. Unsurprisingly, age, number of live births and desire for another child were found to influence use of modern contraception. The number of channels of exposure to FP messaging and visits from a health worker in the past year were significantly related to modern contraceptive use, even after controlling for socio-economic factors. This indicates the importance of continued FP communication and outreach programs in Kinshasa.

Consistent with the lower contraceptive use among women in union in the military population, they are also less likely to report that their last birth was intended (45.5%) as compared to their non-military counterparts (58.1%). It is noteworthy that in both populations more than four in 10 women reported that their last birth was unintended, suggesting the need for intensifying FP programming in Kinshasa.

The study design does not allow for an assessment of the effectiveness of the interventions in the military population. However, some might conclude from the results of this study that the interventions among the military population had little impact, since both knowledge and use of contraception increased significantly over time in **both** the military and non-military populations.

However, several findings point to an alternative interpretation. The women in the military "started from behind" in both knowledge and use. Data from the 2016 survey (not collected among the military in 2019) showed the military population to have a lower economic level than the non-military population. Because the military camps had been considered off-limits to specialized FP programs prior to 2016, it is not surprising that women in this population had lower levels of knowledge and use, compounded by a lower economic level. By 2019, the women in the military population reported higher exposure to FP communication than their non-military counterparts, a finding that would have been unheard of prior to 2016. Moreover, exposure to FP messaging was positively associated with contraceptive use in both populations. Another change in attitudes among married women in the military was the significant difference between surveys in the desire for another child (from 45.5% to 31.6%) among the military, whereas among the non-military population, it remained unchanged. Although the gap in MCPR remained between the two populations, one can plausibly conclude that programmatic activity contributed to an increased MCPR, at a pace similar to the non-military population. Findings from these two quantitative studies underscore the need for additional research using qualitative methods to better understand the dynamics on contraceptive use in this population.

This analysis has several limitations. First, the data from the military and non-military were drawn from different studies, although the methodologies for the two were highly similar. Despite the selection of the same units of residence among the military population for the two surveys, the respondent populations differed on several characteristics: mean age, husband living with wife, and several reproductive history variables. This fact could introduce some

degree of bias. Second, in the 2019 survey, the sample was limited to women married or in union. Since MCPR is often based on women married or in union, this change in selection criterion does not have a large effect on our analysis. However, we have no data on young unmarried sexually active women in the military population, whereas data from the PMA surveys among the non-military women repeated show higher levels of modern contraceptive use among this group compared to those married/in union [2]. Third, the military study in 2019 did not include data needed to measure wealth quintile, which restricted our efforts to include this variable in the multivariate analysis. Fourth, the results are not generalizable beyond the military population in Kinshasa, although they are among the very limited data available on contraceptive use among military populations in sub-Saharan Africa.

Nationally representative studies often exclude the military as being an "institutionalized population. The current experience of conducting a survey in military camps yielded valuable lessons learned. In the current case, the military personnel associated with the study recommended that interviewers and supervisors be selected from the military population to be interviewed, to enhance the acceptability of their asking sensitive questions to women in the military health zones. Similarly, the personnel trained to provide contraceptive services were from the military health zones. These two cross-sectional studies point to the feasibility of conducting research among a military population; in fact, it was the persistent advocacy of a high-level military official and co-author on this article (HNE) that contributed to the decision to conduct the 2019 survey.

At the same time, these initial surveys of the military population in Kinshasa increase our awareness of how much that remains to be learned about contraceptive dynamics in this population. Key questions to be pursued in future research include the following. Does the deployment of the husband influence the choice of method by the wife? Does deployment have a multiplier effect in increasing contraceptive knowledge and use in other military camps? In short, more in-depth insights into contraceptive dynamics in this population will allow for more tailored FP programming, adapted to the needs of this special population.

## Supporting information

**S1 File. 2016 military survey questionnaire, English.**
(DOCX)

**S2 File. 2016 military survey questionnaire, French.**
(DOCX)

**S3 File. 2020 military survey questionnaire, English.**
(DOCX)

**S4 File. 2020 military survey questionnaire, French.**
(DOCX)

**S1 Table. Full regression output.**
(DOCX)

**S1 Dataset. Military survey data.**
(DTA)

## Acknowledgments

We lament the passing of Dr. Patrick Kayembe, dear friend and colleague, before the submission of the final version of this manuscript. Katherine H LaNasa accepts responsibility for the integrity and validity of the data collected and analyzed.

## Author Contributions

**Conceptualization:** Pierre Z. Akilimali, Henri Engale Nzuka, Angéle Mavinga Wumba, Patrick Kayembe.

**Formal analysis:** Pierre Z. Akilimali, Janna Wisniewski.

**Funding acquisition:** Jane T. Bertrand.

**Investigation:** Pierre Z. Akilimali.

**Methodology:** Pierre Z. Akilimali.

**Project administration:** Henri Engale Nzuka, Angéle Mavinga Wumba, Patrick Kayembe.

**Resources:** Henri Engale Nzuka, Angéle Mavinga Wumba, Patrick Kayembe.

**Supervision:** Jane T. Bertrand.

**Validation:** Pierre Z. Akilimali.

**Visualization:** Pierre Z. Akilimali.

**Writing – original draft:** Katherine H. LaNasa, Jane T. Bertrand.

**Writing – review & editing:** Katherine H. LaNasa, Janna Wisniewski, Jane T. Bertrand.

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
