## [Decision Letter · Decision Letter 0]

2 Mar 2021

PONE-D-21-04034

Closing the Gap: Contraceptive dynamics in military and non-military populations of Kinshasa, DRC, 2016-2020

PLOS ONE

Dear Dr. Schulze,

Thank you for submitting your manuscript to PLOS ONE. After careful consideration, we feel that it has merit but does not fully meet PLOS ONE’s publication criteria as it currently stands. Therefore, we invite you to submit a revised version of the manuscript that addresses the points raised during the review process.

Two of PLOS publication criteria are that experiments, statistics, and other analyses are performed to a high technical standard and are described in sufficient detail and methods must be described in sufficient detail for another researcher to reproduce the experiments described; and that conclusions are presented in an appropriate fashion and are supported by the data.

Based on the comments of the reviewers both aspects need to be improved. In addition, regarding the data and the methods there are more questions:

1. The distribution by age in 2016 and 2020 shows that the military population is clearly not the same. The age distribution is completely different. More background information should be given on why this is so. Note also that, not being the same population, the interpretation of the difference-in-difference is more troublesome, since we might be testing for differences between the two different military populations of 2016 and 2020, than changes over time or "closing the gap" as the title suggests. This has to be acknowledged as a major limitation, and bore in mind when explaining the results. You do not comment, and it is telling regarding the difference in populations, that despite the less children ever born (in line with being younger in 2020), they have more child deaths. I know PMA2020 does not give data, but what happened? Was there some epidemic mortality between 2016 and 2020 that might explain this, or is it rather a hint at a more deprived background population? Also, as the reviewers suggest, differences are not significant in any of the 2 measures. Regarding this, please rearrange tables so that the interaction comes right after the two main effects, population and year.

2. There are no details on how the pooling of the data from the 4 surveys was carried out, in particular sample weights, ... See also the specific questions on the comparability by the reviewers.

3. It is said that data is available without restriction. However, data from the military surveys is not attached to the submission and there is no data repository given. Data availability is a requirement for publication at PLOS ONE.

We look forward to receiving your revised manuscript.

Kind regards,

José Antonio Ortega, Ph.D.

Academic Editor

PLOS ONE

Journal Requirements:

3. In your Methods section, please provide additional information about the participant recruitment method and the demographic details of your participants. Please ensure you have provided sufficient details to replicate the analyses such as:

- the recruitment date range (month and year)

- a description of any inclusion/exclusion criteria that were applied to participant recruitment

- a statement as to whether your sample can be considered representative of a larger population.

4. We noted in your submission details that a portion of your manuscript may have been presented or published elsewhere.

"The baseline survey results, collected in 2016, were reported in the 2018 article published in BMJ Open, entitled “Differences in family planning outcomes between military and general populations in Kinshasa, Democratic Republic of the Congo: a cross-sectional analysis” by Akilimali et al. Our article provides the follow-up results from 2020 and assesses the change over time in family planning-related indicators between the two survey rounds."

Reviewers' comments:

Reviewer's Responses to Questions

**Comments to the Author**

1. Is the manuscript technically sound, and do the data support the conclusions?

Reviewer #1: Partly

Reviewer #2: Partly

2. Has the statistical analysis been performed appropriately and rigorously? 

Reviewer #1: Yes

Reviewer #2: Yes

3. Have the authors made all data underlying the findings in their manuscript fully available?

Reviewer #1: Yes

Reviewer #2: Yes

4. Is the manuscript presented in an intelligible fashion and written in standard English?

Reviewer #1: No

Reviewer #2: Yes

5. Review Comments to the Author

Reviewer #1: Peer Review:

PONE-D-21-04034

Closing the Gap: Contraceptive dynamics in military and non-military populations of Kinshasa, DRC, 2016-2020

PLOS ONE

Overview:

This is an interesting study examining the impact of changes in family planning (FP) programing between 2016 and 2020 on contraceptive use among women living on military camps in Kinshasa, Democratic Republic of Congo (DRC). This is an important topic of study because military members and their families in DRC and other sub-Saharan countries are an understudied population with higher rates of unintended pregnancies and HIV infection than the surrounding population. Implementing effective reproductive health programing in the communities where these women live and, in the military-specific clinics where they seek care has the potential improve the health of this higher-risk population. This study could assist with identifying effective family planning programing for female servicemembers or women living in military families. However, I think several issues need to be addressed by the authors to assist readers with interpreting and applying the conclusions from their research.

Major Concerns:

1. I suggest the authors limit their description of results and conclusions to statements that are supported by their study data.

Lines 40-42: “ Although the change in modern contraception use did not significantly differ between the populations, the increase in use among military women (15.0% to 24.2%) was observably larger among non-military women (23.1% to 29.7%).”

Lines 50-52: “and points to the potential value of developing FP programming adapted to the unique needs of the military population.”

Lines 299-303: “The change in use of modern contraception between survey years did not significantly differ between military and non-military women. However, the percentage point increase in use of modern contraception from 2016 to 2020 (Table 3) among military women (9.2 percentage points) was observably larger than the increase among non-military women (6.6 percentage points).

Women on military camps in Kinshasa, DRC in 2016 reported higher rates of exposure to FP programing than women in the surrounding community. However, women on military camps had a lower rate of contraception use and a higher rate of unplanned pregnancy. Between 2016 and 2020 exposure to FP programing increased among women living on military camps and decreased among women living in the surrounding community. There was an increase in use of contraception, modern contraception, and use of their preferred contraceptive method among women living on military camps between 2016 and 2020. However, these changes were not statistically different from the changes seen among women in the surrounding community. This suggests that the FP programing offered in the DRC military camps prior to 2016 and the increase in this programing between 2016 and 2020 were ineffective at improving contraceptive outcomes or at least this study had insufficient power to detect the impact of the increased FP programing.

The authors have demonstrated that they can gather reproductive health data among females living in military camps and demonstrated the importance of gathering outcomes data in determining if your intervention is having the desired outcome. A discussion of what they did and why they thought it did not have the desired outcome would be interesting.

2. I think a further discussion of the survey tools used in this study for the intervention group (military) and control group (PMA) would be useful in interpreting the results of this study.

This study used data from a military specific survey conducted with a sample of women living on one of the 17 military camps in Kinshasa, DRC. The authors used data from the annual Performance Monitoring and Accountability Project (PMA), collected in 2016 and 2020, to represent the experience of the non-military population in Kinshasa. The authors do a good job discussing the sampling method used by each survey.

However, as the authors note, “the data from the military and non-military were drawn from different studies, although the methodologies for the two were highly similar. Nonetheless, this fact could introduce some degree of bias.” (lines 369-371) I would like further discussion of the comparability of these two surveys in terms of the specific questions used, how the surveys were administered once a household was selected, and who was interviewed. I am confused by the current descriptions:

For example, it is unclear to me if the sample was restricted to married women in both surveys, married or in union in both surveys, or married in the military sample and married or in union in the PMA.

PMA Description:

Lines 127-131: “At each selected house, the head of the household is first asked to complete a household survey, then all resident women of reproductive age (15-49 years) are interviewed. The PMA female survey, which is conducted with only female interviewers, consists of basic demographic information and in-depth information on fertility history and preferences and use of contraception.”

Lines 135-137: “Our analysis is restricted to only married women, so in total, 1,288 women from the 2016 PMA survey and 1,085 women from the 2020 PMA survey are included in this study.”

Military Survey Description:

Lines 143-144: All married women of reproductive age (15-49 years) within a selected household were interviewed.

Sociodemographic Profile Description of Sample:

Lines 207-208: “All results are based on women married or in union aged 15-49 years old.”

Lines 371-377: “Second, in the 2020 survey, the sample size for women in the military was reduced, and the sample was limited to women married or in union. Since mCPR is often based on women married or in union, this change in selection criterion does not have a large effect on our analysis. However, we have no data on young unmarried sexually active women in the military population, whereas data from the PMA surveys among the non-military women repeated show higher levels of modern contraceptive use among this group compared to those married/in union.”

The authors also state that the PMA female survey was “conducted with only female interviewers” (line 125) However, they do not say how the interview was delivered (for example computer, verbally, pen and paper in the woman’s native language) and they do not say how the military interview was delivered at all.

In the measures section (lines 152-183), the authors discuss the topics covered in the PMA and military survey, including topics that were only covered in some of the surveys. However, the authors never say if the exact same questions were asked in the two surveys and if they were presented in the same way. The questions asked and the method used to deliver the questions can have effects on the data obtained.

3. Comparability of the military and non-military groups

The authors describe the difference between the 2016 and 2020 within the military and non-military samples. It would be useful to also see an assessment of differences between the military and non-military samples in 2016 and 2020. It appears that women in the PMA survey were more likely to have planned their last childbirth, want another child, and currently have the method of contraception they desire. I would like the authors to explore the differences between the samples further and discuss how these differences between the intervention and control group, in regard to childbirth intentions and desires, impact their findings.

4. Further description of the intervention on the military camps

In the introduction section the authors describe the intensification of FP for the military population.

Lines 88-95: “The military population in Kinshasa has benefited from intensified FP programming, starting in 2016. Nursing personnel in 10 health centers within 14 military camps received training in FP; messaging to promote FP has highlighted military families (such as a billboard posted near the entrance of the main military camp, Camp Kokolo, which shows the typical “healthy family with well-spaced children” with the father in uniform (see Figure 1). Since 2016, the AcQual project has established community-based distribution including quarterly mini-campaigns (outreach activities with counseling and service provision of five contraceptive methods) in a community-setting, often near a referral health center.”

The authors could consider moving this information to the methods sections. The paper would also benefit from a more detailed discussion what interventions were offered and how this impacted outcomes. For example, did outcomes differ between the 10 camps which obtained new nursing personnel and the 4 camps that did not?

Based on the 2016 data presented by the authors, it appears that women in military camps had a higher exposure to channels of FP information and more visits by a health worker than women in the surrounding community prior to the intervention. (Table 2) In their multivariable analysis (Table 4), the authors identify these two factors as significant predictors of modern contraceptive use. However, modern contraceptive use and ability to obtain the contraceptive you desired were higher in the population who were receiving less of these factors in 2016. Why do the authors think this is occurring? How did the interventions between 2016 and 2020 differ from what was being offered prior to 2016?

5. Selection of outcome measures for the multivariable analyses

I agree with the authors that prevention of unintended pregnancies is an important outcome when assessing contraceptive interventions. However, I would like them to discuss why they attempted to predict prior unintended pregnancies with current exposure to current FP programing. I can understand how a prior unintended or undesired pregnancy could influence current contraceptive behavior or the saliency of FP messaging, but I do not comprehend the reverse. I would like the authors to discuss this decision further or reconsider this analysis.

6. Selection of explanatory variables

I would like the authors to provide further discussion of why they are not including desire for a future child in their analysis of modern contraceptive use. I suspect that women who do not want a future child are more likely to seek out and use more effective contraceptive methods. The authors could also consider using experience of a prior unintended pregnancy, possibly occurring because of failure of their prior contraceptive method, as an explanatory variable for use of modern contraception.

Reviewer #2: This study compares samples of married women in military and non-military populations in Kinshasa, DRC, with regards to contraceptive use, knowledge, and exposure to FP information. This is important work in an important population, but I believe that authors could enhance the paper by including more context and direct discussion of future research directions. Please see comments below for more details:

In general, I would suggest being careful with regards to the description of the analysis. There are places where the authors state that the information found in tables 1-3 are being “compared” between military and non-military populations (like in the listed objectives and elsewhere). However, it does not appear that the groups are being directly compared statistically, but rather tests are being run within each group. It does seem like this is accurately described in the analysis as being analyzed “…among military and non-military women.” Did the authors consider comparing the two groups directly on the variables in tables 1-3?

ABSTRACT: METHODS: Should be clearly stated that two different survey of military and non-military populations were completed.

INTRO: Are there any specific norms or policies with in military culture that might help interpret findings? This is hinted at in the abstract where it states that the results point “to the potential value of developing FP programming adapted to the unique needs of the military population,” and perhaps these unique needs are not yet known, but it would be helpful to suggest interpretations based on context or identify as a future direction research to uncover unique needs of military populations.

MEASURES: Other than the few instances where questions were only asked of military populations, or were only asked at one time point in the non-military survey, were the items identical? Did they use the same terminology and response options? If so, this should be explicitly stated. If not, this could be considered a limitation in trying to directly compare the two samples.

ANALYSIS: Perhaps it is because of the misplaced parenthesis on page 10, line 196, but the last few sentences of the analysis section are dense and difficult to read.

TABLES: Would help the reader to be clear that numbers in tables 1-3 are percentages unless specified as a mean (e.g., age). Also, it is not entirely clear why some significant results are bolded and others are not.

DISCUSSION: Some interesting findings I was hoping that discussion might address: 1) why were fewer military women living with husband in 2020 vs. 2016? 2) Why were women across the board less knowledgeable about female condoms? 3) What happened to knowledge of rhythm method in military women? 4) I noticed that the exposure sources included TV, radio, and magazine/newspaper. How is internet penetration within Kinshasa? Are these sources outdated? Are FP campaigns conducted by internet? Could less use of “outdated” sources account for drop in exposure in non-military? 5) Why are more births unintended in military populations?

It is noted on line 329 that deployment of men may influence contraceptive choice; could the authors elaborate on how?

The authors seem to note that younger unmarried women in the PMA surveys showed higher levels of modern contraceptive use. Is there a reference that can be added for these findings?

I think the last point about feasibility is important and an understated finding of this study. Perhaps that point could be highlighted along with explicit future research directions.

6. PLOS authors have the option to publish the peer review history of their article (what does this mean?). If published, this will include your full peer review and any attached files.

Reviewer #1: **Yes: **Timothy A. Roberts, MD MPH

Reviewer #2: No

---

## [Author Response · Author response to Decision Letter 0]

28 Apr 2021

Authors’ replies to the editors/reviewers’ comments: ONE-D-21-04034

(previous title) Closing the Gap: Contraceptive dynamics in military and non-military populations of Kinshasa, DRC, 2016-2020

(new title) The Gap in Contraceptive Knowledge and Use between the Military and Non-military populations of Kinshasa, DRC, 2016-2019

Note: in this document, all comments from the editors/reviewers are shown in bold (non-italics). All responses from the authors are shown in non-bold italics. In addition, we would like to draw attention to the change in the data of the second survey in our revised manuscript. Based on the actual dates of data collection, the surveys in both the military and non-military populations are much closer to a three- than a four-year interval. (Because the data collection for the second survey in both cases extended into 2020, we had opted for simplicity to describe it as a comparison of 2016 and 2019. We now specify that there was a three-year interval and (again for simplicity) have labeled the second surveys as occurring in 2019, although the exact dates are given in the methods section.

Two of PLOS publication criteria are that experiments, statistics, and other analyses are performed to a high technical standard and are described in sufficient detail and methods must be described in sufficient detail for another researcher to reproduce the experiments described; and that conclusions are presented in an appropriate fashion and are supported by the data.

Based on the comments of the reviewers both aspects need to be improved. In addition, regarding the data and the methods there are more questions:

1. The distribution by age in 2016 and 2020 shows that the military population is clearly not the same. The age distribution is completely different. More background information should be given on why this is so. Note also that, not being the same population, the interpretation of the difference-in-difference is more troublesome, since we might be testing for differences between the two different military populations of 2016 and 2020, than changes over time or "closing the gap" as the title suggests. 

Reply: we did not do difference of differences analysis in Tables 1-3 but did in Table 4.

This has to be acknowledged as a major limitation, and bore in mind when explaining the results. You do not comment, and it is telling regarding the difference in populations, that despite the less children ever born (in line with being younger in 2020), they have more child deaths. I know PMA2020 does not give data, but what happened? Was there some epidemic mortality between 2016 and 2020 that might explain this, or is it rather a hint at a more deprived background population? Also, as the reviewers suggest, differences are not significant in any of the 2 measures. Regarding this, please rearrange tables so that the interaction comes right after the two main effects, population and year.

Reply: we have added several points to the paper to explain these differences. In the methods section, we explain that the units of residence remained fixed across surveys in the military population but not in the non-military population (PMA). In the Discussion, we further describe the military population as being more mobile (due to deployments) and economically more deprived. As for the difference in number of child deaths, we do not have a ready explanation. However, we have added the following to the limitations section:

• First, the data from the military and non-military were drawn from different studies, although the methodologies for the two were highly similar. Despite the selection of the same units of residence among the military population for the two surveys, the respondent populations differed on several characteristics: mean age, husband living with wife, and several reproductive history variables. This fact could introduce some degree of bias.

We have rearranged Table 4 so that the interaction comes after the two main effects, population and year. To mitigate the impact of differences between groups, we have controlled for individual characteristics in that model.

2. There are no details on how the pooling of the data from the 4 surveys was carried out, in particular sample weights, ... See also the specific questions on the comparability by the reviewers.

Reply: We have pooled the data to include both waves and then created a variable for wave; we have used the weight for each individual wave/survey group.

3. It is said that data is available without restriction. However, data from the military surveys is not attached to the submission and there is no data repository given. Data availability is a requirement for publication at PLOS ONE.

Reply: The data for the two military surveys have been posted in Dryad. The citation is:

Akilimali, Pierre et al. (2021), The gap in knowledge and contraceptive use in military and non-military populations of Kinshasa, DRC, 2016-2020, Dryad, Dataset. https://doi.org/10.5061/dryad.k0p2ngf85

Reply: our resubmission conforms to these instructions.

Reply: we do not intend to make changes in our financial disclosure.

Reply: this point is not applicable.

Journal Requirements:

Reply: our resubmission conforms to these instructions.

Reply: we have submitted a copy of the questionnaires used in the military surveys as Supporting Information. The questionnaires for PMA2020 and PMA are available from their websites. 

3. In your Methods section, please provide additional information about the participant recruitment method and the demographic details of your participants. Please ensure you have provided sufficient details to replicate the analyses such as:

- the recruitment date range (month and year)

- a description of any inclusion/exclusion criteria that were applied to participant recruitment

- a statement as to whether your sample can be considered representative of a larger population. 

Reply: the dates for data collection for the PMA surveys already appeared in the manuscript. We have added the month/year for the two surveys among the military in the Methods section.

We have described the sampling procedures and provided additional detail on the sampling procedures and data collection (shown in red font below) as follows:

The second data source is a separate survey among the military population, conducted at an interval of approximately three years: from Nov 19-Dec 12, 2016 and from Dec 29, 2019-Jan 21, 2020. (Again, for simplicity, we have labeled the latter as the “2019 survey.”) To capture a representative sample of this population, a similar two-stage cluster design was used. Ten military camps were randomly selected, using selection probability based on population size, out of the 17 camps in Kinshasa. The selected camps were then divided in EAs, and one EA was randomly selected in each of the 10 camps. Following the PMA format, the households in each selected EA were listed and 33 household were randomly selected. The same units of residence were used in both military surveys, although the residents could have changed. In the 2016 military survey, all married women of reproductive age (15-49 years) living in a selected household were interviewed; in the 2019 survey only women 15-49 married or in union were interviewed, since we planned to limit this analysis to women married of living in union. The cases available for analysis were n=229 in 2016, n=357 in 2019. Based on the sampling techniques used, the data can be considered generalizable to women in the military populations and the non-military populations in Kinshasa. 

The content and procedures for the PMA and military surveys were highly similar on the following points. The local study directors were the same for both surveys. The vast majority of the questions in the military surveys were taken verbatim from the PMA questionnaire, with the same terminology and response options, except for a few additional questions not found in the PMA questionnaire. The questions were administered by trained female interviewers in French or Lingala (the local language). Data were collected in both rounds and both populations using tablets or smartphones programmed with ODK. 

4. We noted in your submission details that a portion of your manuscript may have been presented or published elsewhere.

"The baseline survey results, collected in 2016, were reported in the 2018 article published in BMJ Open, entitled “Differences in family planning outcomes between military and general populations in Kinshasa, Democratic Republic of the Congo: a cross-sectional analysis” by Akilimali et al. Our article provides the follow-up results from 2020 and assesses the change over time in family planning-related indicators between the two survey rounds."

Reply: The previous article was published in a peer-reviewed journal: 

Akilimali P, Anglewicz P, Engale H, Kurhenga G, Hernandez J, Kayembe P, et al. Differences in family planning outcomes between military and general populations in Kinshasa, Democratic Republic of the Congo: a cross-sectional analysis. BMJ Open. 2018;8(e022295). doi: 10.1136/ bmjopen-2018-022295.

As stated in the cover letter, the current manuscript analyzes the change over time in contraceptive use in the military and non-military population. It would not be possible to conduct this analysis without incorporating the baseline data for the two populations. In short, the objective of this second article differs from the first in that it measures change over time.

Reply: the authors have added this variable to table 1 for both 2016 and 2019 (military only) and removed “(data not shown).”

Reply: our resubmission conforms to these instructions.

Reviewer #1: Peer Review:

PONE-D-21-04034

Closing the Gap: Contraceptive dynamics in military and non-military populations of Kinshasa, DRC, 2016-2020

Overview:

This is an interesting study examining the impact of changes in family planning (FP) programing between 2016 and 2020 on contraceptive use among women living on military camps in Kinshasa, Democratic Republic of Congo (DRC). This is an important topic of study because military members and their families in DRC and other sub-Saharan countries are an understudied population with higher rates of unintended pregnancies and HIV infection than the surrounding population. Implementing effective reproductive health programing in the communities where these women live and, in the military-specific clinics where they seek care has the potential improve the health of this higher-risk population. This study could assist with identifying effective family planning programing for female servicemembers or women living in military families. However, I think several issues need to be addressed by the authors to assist readers with interpreting and applying the conclusions from their research.

Major Concerns:

1. I suggest the authors limit their description of results and conclusions to statements that are supported by their study data.

Lines 40-42: “ Although the change in modern contraception use did not significantly differ between the populations, the increase in use among military women (15.0% to 24.2%) was observably larger among non-military women (23.1% to 29.7%).” 

Reply: Consistent with the reviewers’ comments, we have largely rewritten the abstract and removed the point about the difference being “observably larger.” Also, we have changed the title to read: “The Gap in Contraceptive Knowledge and Use between the Military and Non-military populations of Kinshasa, DRC, 2016-2019.”

Lines 50-52: “and points to the potential value of developing FP programming adapted to the unique needs of the military population.”

Reply: The authors have provided additional text in the Discussion on the interpretation of the results and the importance of additional research, especially qualitative, to better understand the unique needs of the military population. 

Lines 299-303: “The change in use of modern contraception between survey years did not significantly differ between military and non-military women. However, the percentage point increase in use of modern contraception from 2016 to 2020 (Table 3) among military women (9.2 percentage points) was observably larger than the increase among non-military women (6.6 percentage points). 

Reply: The authors have removed the sentence: “However, the percentage point increase in use of modern contraception from 2016 to 2019 (Table 3) among military women (9.2 percentage points) was observably larger than the increase in the non-military popular women (6.6 percentage points). In the next paragraph we have added the sentence: “Although the amount of change between the two populations was not significantly different, it is notable that by 2019, the MCPR for women in the military population (24.2%) had reached the MCPR level of non-military women four years earlier (23.1%). 

Women on military camps in Kinshasa, DRC in 2016 reported higher rates of exposure to FP programing than women in the surrounding community. However, women on military camps had a lower rate of contraception use and a higher rate of unplanned pregnancy. Between 2016 and 2020 exposure to FP programing increased among women living on military camps and decreased among women living in the surrounding community. There was an increase in use of contraception, modern contraception, and use of their preferred contraceptive method among women living on military camps between 2016 and 2020. However, these changes were not statistically different from the changes seen among women in the surrounding community. This suggests that the FP programing offered in the DRC military camps prior to 2016 and the increase in this programing between 2016 and 2020 were ineffective at improving contraceptive outcomes or at least this study had insufficient power to detect the impact of the increased FP programing. 

Reply: The authors have further commented on the differences between the populations, including in programming prior to 2016. Also, in the discussion we address the reviewer’s claim that the programming was ineffective, if the increases between the military and non-military were not significantly different:

The study design does not allow for an assessment of the effectiveness of the interventions in the military population. However, some might conclude from the results of this study that the interventions among the military population had little impact, since both knowledge and use of contraception increased significantly over time in both the military and non-military populations. 

However, several findings point to an alternative interpretation. The women in the military “started from behind” in both knowledge and use. Data from the 2016 survey (not collected among the military in 2019) showed the military population to have a lower economic level than the non-military population. Because the military camps had been considered off-limits to specialized family planning programs prior to 2016, it is not surprising that women in this population had lower levels of knowledge and use, compounded by a lower economic level. By 2019, the women in the military population reported higher exposure to FP communication than their non-military counterparts, a finding that would have been unheard of prior to 2016. Moreover, exposure to FP messaging was positively associated with contraceptive use in both populations. Another change in attitudes among married women in the military was the significant decrease in the desire for another child (from 45.5% to 31.6%) among the military, whereas among the non-military population, it remained unchanged. Although these interventions (and other factors) were not sufficient to close the gap in MCPR between the two populations, one can plausibly conclude that they contributed to allowing the military population to continue to improve on MCPR, at a pace similar to the non-military population. Findings from these two quantitative studies underscore the need for additional research using qualitative methods to better understand the dynamics on contraceptive use in this population.

The authors have demonstrated that they can gather reproductive health data among females living in military camps and demonstrated the importance of gathering outcomes data in determining if your intervention is having the desired outcome. A discussion of what they did and why they thought it did not have the desired outcome would be interesting.

Reply: See the text directly in reply to the question directly above. 

2. I think a further discussion of the survey tools used in this study for the intervention group (military) and control group (PMA) would be useful in interpreting the results of this study.

This study used data from a military specific survey conducted with a sample of women living on one of the 17 military camps in Kinshasa, DRC. The authors used data from the annual Performance Monitoring and Accountability Project (PMA), collected in 2016 and 2020, to represent the experience of the non-military population in Kinshasa. The authors do a good job discussing the sampling method used by each survey.

However, as the authors note, “the data from the military and non-military were drawn from different studies, although the methodologies for the two were highly similar. Nonetheless, this fact could introduce some degree of bias.” (lines 369-371) I would like further discussion of the comparability of these two surveys in terms of the specific questions used, how the surveys were administered once a household was selected, and who was interviewed. I am confused by the current descriptions:

For example, it is unclear to me if the sample was restricted to married women in both surveys, married or in union in both surveys, or married in the military sample and married or in union in the PMA.

PMA Description:

Lines 127-131: “At each selected house, the head of the household is first asked to complete a household survey, then all resident women of reproductive age (15-49 years) are interviewed. The PMA female survey, which is conducted with only female interviewers, consists of basic demographic information and in-depth information on fertility history and preferences and use of contraception.”

Lines 135-137: “Our analysis is restricted to only married women, so in total, 1,288 women from the 2016 PMA survey and 1,085 women from the 2020 PMA survey are included in this study.”

Military Survey Description:

Lines 143-144: All married women of reproductive age (15-49 years) within a selected household were interviewed.

Reply: we have responded to a similar question by the other reviewer above and have shown (in red font) the changes made to the Methods section to clarify these points.

Sociodemographic Profile Description of Sample: 

Lines 207-208: “All results are based on women married or in union aged 15-49 years old.”

Lines 371-377: “Second, in the 2020 survey, the sample size for women in the military was reduced, and the sample was limited to women married or in union. Since mCPR is often based on women married or in union, this change in selection criterion does not have a large effect on our analysis. However, we have no data on young unmarried sexually active women in the military population, whereas data from the PMA surveys among the non-military women repeated show higher levels of modern contraceptive use among this group compared to those married/in union.”

The authors also state that the PMA female survey was “conducted with only female interviewers” (line 125) However, they do not say how the interview was delivered (for example computer, verbally, pen and paper in the woman’s native language) and they do not say how the military interview was delivered at all.

In the measures section (lines 152-183), the authors discuss the topics covered in the PMA and military survey, including topics that were only covered in some of the surveys. However, the authors never say if the exact same questions were asked in the two surveys and if they were presented in the same way. The questions asked and the method used to deliver the questions can have effects on the data obtained.

Reply: we have added details to the Methods section to address these questions.

3. Comparability of the military and non-military groups

The authors describe the difference between the 2016 and 2020 within the military and non-military samples. It would be useful to also see an assessment of differences between the military and non-military samples in 2016 and 2020. It appears that women in the PMA survey were more likely to have planned their last childbirth, want another child, and currently have the method of contraception they desire. I would like the authors to explore the differences between the samples further and discuss how these differences between the intervention and control group, in regard to childbirth intentions and desires, impact their findings.

Reply: we wouldn’t characterize the two groups as intervention and control, but rather two naturally occurring groups in the population of Kinshasa. Table 1 presents the differences in the two groups for both surveys, and in the results section, we comment briefly on these differences. We have added further description of the military population as being of lower socio-economic status in the Discussion. Although we did find some significant differences between the military population and on the two surveys, we control for socio-economic characteristics in the logistic regression in Table 4. 

4. Further description of the intervention on the military camps

In the introduction section the authors describe the intensification of FP for the military population.

Lines 88-95: “The military population in Kinshasa has benefited from intensified FP programming, starting in 2016. Nursing personnel in 10 health centers within 14 military camps received training in FP; messaging to promote FP has highlighted military families (such as a billboard posted near the entrance of the main military camp, Camp Kokolo, which shows the typical “healthy family with well-spaced children” with the father in uniform (see Figure 1). Since 2016, the AcQual project has established community-based distribution including quarterly mini-campaigns (outreach activities with counseling and service provision of five contraceptive methods) in a community-setting, often near a referral health center.”

The authors could consider moving this information to the methods sections. 

Reply: the authors have moved the paragraph above to the methods section, as suggested.

The paper would also benefit from a more detailed discussion what interventions were offered and how this impacted outcomes. For example, did outcomes differ between the 10 camps which obtained new nursing personnel and the 4 camps that did not?

Reply: the sample size is too small to allow for this type of more in-depth analysis. 

Based on the 2016 data presented by the authors, it appears that women in military camps had a higher exposure to channels of FP information and more visits by a health worker than women in the surrounding community prior to the intervention. (Table 2) In their multivariable analysis (Table 4), the authors identify these two factors as significant predictors of modern contraceptive use. However, modern contraceptive use and ability to obtain the contraceptive you desired were higher in the population who were receiving less of these factors in 2016. Why do the authors think this is occurring?

Reply: As noted above, the authors have added the following text to the Discussion: 

The study design does not allow for an assessment of the effectiveness of the interventions in the military population. However, some might conclude from the results of this study that the interventions among the military population had little impact, since both knowledge and use of contraception increased significantly over time in both the military and non-military populations. 

However, several findings point to an alternative interpretation. The women in the military “started from behind” in both knowledge and use. Data from the 2016 survey (not collected among the military in 2019) showed the military population to have a lower economic level than the non-military population. Because the military camps had been considered off-limits to specialized family planning programs prior to 2016, it is not surprising that women in this population had lower levels of knowledge and use, compounded by a lower economic level. By 2019, the women in the military population reported higher exposure to FP communication than their non-military counterparts, a finding that would have been unheard of prior to 2016. Moreover, exposure to FP messaging was positively associated with contraceptive use in both populations. Another change in attitudes among married women in the military was the significant decrease in the desire for another child (from 45.5% to 31.6%) among the military, whereas among the non-military population, it remained unchanged. Although these interventions (and other factors) were not sufficient to close the gap in MCPR between the two populations, one can plausibly conclude that they contributed to allowing the military population to continue to improve on MCPR, at a pace similar to the non-military population. Findings from these two quantitative studies underscore the need for additional research using qualitative methods to better understand the dynamics on contraceptive use in this population.

 How did the interventions between 2016 and 2020 differ from what was being offered prior to 2016?

Reply: we have added the following text to the Methods section, under “Setting.”

The city of Kinshasa consists of 35 health zones, including three which are highly rural and distant from the center of the city (thus, atypical). The military health zone of Kokolo consists of 15 non-adjacent “camps,” which operate as isolated settlements within the boundaries of the highly populated areas of the city. Fourteen of the 15 camps have a health facility, operated by the military for the residents of the camp. 

Prior to the 2016 survey, some of the health facilities within Kokolo sporadically had contraceptives available, but stockouts were frequent and the range of methods was limited. No group education sessions or other types of community awareness raising took place. However, as of 2016, the military established a Coordination Unit for Reproductive Health for the military camps. Numerous activities were conducted to reinforce the provision of FP services: training of clinical personnel within the military-run health facilities, training of community-health workers to serve as community-based distributors (CBD), regular resupply of facilities and CBD with a full range of contraceptives, provision of materials needed for the administration of implants and IUDs, mini-campaigns (outreach activities) several times a year that heightened the visibility and improved access to contraceptives, and promotional materials (such as the family planning billboard outside the gates of the camp depicting a military family, shown in Figure 1). 

In the non-military health zones of Kinshasa, family planning service provision varies by health zone, often depending on support from international NGOs; there is no “standard treatment” city-wide. The non-military health zones generally benefit from the same combination of activities as the military health zones, with a less systematic or regular delivery. In short, the programmatic intervention in the military zone starting in 2016 was designed to ensure the same service delivery inputs as many of the non-military health zones were already receiving, albeit in more piecemeal fashion.

5. Selection of outcome measures for the multivariable analyses

I agree with the authors that prevention of unintended pregnancies is an important outcome when assessing contraceptive interventions. However, I would like them to discuss why they attempted to predict prior unintended pregnancies with current exposure to current FP programing. I can understand how a prior unintended or undesired pregnancy could influence current contraceptive behavior or the saliency of FP messaging, but I do not comprehend the reverse. I would like the authors to discuss this decision further or reconsider this analysis.

Reply: We concur with the reviewer that the order of causality is not clear in the case of unintended pregnancy, and we have removed this outcome variable from the analysis. We have removed unintended pregnancy as an outcome in Table 4 and from the results section.

6. Selection of explanatory variables

I would like the authors to provide further discussion of why they are not including desire for a future child in their analysis of modern contraceptive use. I suspect that women who do not want a future child are more likely to seek out and use more effective contraceptive methods. 

Reply: we have included desire for additional child(ren) in the multivariate analysis in Table 4. It is indeed significant, and we have revised the results section as follows:

Several factors were found to be associated with modern contraceptive use. Unsurprisingly, age, number of live births and desire for another child were found to influence use of modern contraception. The number of channels of exposure to FP messaging and visits from a health worker in the past year were significantly related to modern contraceptive use, even after controlling for socio-economic factors. This indicates the importance of continued FP communication and outreach programs in Kinshasa.

The authors could also consider using experience of a prior unintended pregnancy, possibly occurring because of failure of their prior contraceptive method, as an explanatory variable for use of modern contraception.

Reply: since the questionnaire only allows us to capture the previous pregnancy, not any pregnancy, we have opted not to follow up on this suggestion. 

Reviewer #2: This study compares samples of married women in military and non-military populations in Kinshasa, DRC, with regards to contraceptive use, knowledge, and exposure to FP information. This is important work in an important population, but I believe that authors could enhance the paper by including more context and direct discussion of future research directions. Please see comments below for more details:

In general, I would suggest being careful with regards to the description of the analysis. There are places where the authors state that the information found in tables 1-3 are being “compared” between military and non-military populations (like in the listed objectives and elsewhere). However, it does not appear that the groups are being directly compared statistically, but rather tests are being run within each group. It does seem like this is accurately described in the analysis as being analyzed “…among military and non-military women.” Did the authors consider comparing the two groups directly on the variables in tables 1-3?

Reply: The reviewers are correct that in tables 1 to 3, we are comparing each population over two surveys. However, in the multivariate analysis in table 4, we do directly compare change within the two populations.

ABSTRACT: METHODS: Should be clearly stated that two different survey of military and non-military populations were completed.

Reply: we have changed the abstract to read:

Introduction

The objective of this study is to assess change over time in the modern contraceptive prevalence rate (MCPR) and related variables among married women of reproductive age (15-49 years) in the military population in Kinshasa, Democratic Republic of Congo, compared to women in the non-military population, based on cross-sectional surveys in 2016 and 2019. 

Methods

Data among women living in military camps were collected as a special study of contraceptive knowledge, use, and exposure to FP messaging, for comparison to women in the non-military population from the annual PMA2020 survey in the same years. Both used a two-stage cluster sampling design to randomly select participants. This analysis is limited to women married or in union. Bivariate and multivariate analysis was used to compare the military and non-military populations on a number of FP-related indicators.

Results

Knowledge of modern contraceptive methods increased significantly between 2016 and 2019 in both populations, although the military women were still less knowledgeable than their non-military counterparts. Similarly, modern contraceptive use increased significantly in both populations, but the military women continued to trail their non-military counterparts on MCPR (24.2% versus 29.7%.) Among contraceptive users in both populations, the implant was the leading method. Multivariate analysis showed no significant difference in the amount of increase in MCPR for the two populations, although by 2019 the women in the military population had caught up to where the non-military population was as of 2016. Among the military population, exposure to multiple channels of FP messaging increased, while exposure among the non-military population decreased.

Conclusions

This study demonstrates the feasibility and importance of collecting data in military camps for better understanding contraceptive dynamics among this specialized population.

INTRO: Are there any specific norms or policies within military culture that might help interpret findings? This is hinted at in the abstract where it states that the results point “to the potential value of developing FP programming adapted to the unique needs of the military population,” and perhaps these unique needs are not yet known, but it would be helpful to suggest interpretations based on context or identify as a future direction research to uncover unique needs of military populations.

Reply: If there is a norm or policy that relate to the military camps, it is the mindset that one gets what the military provides (in terms of housing or healthcare), without the expectation or aspiration to more. As noted above, the level of living in military camps is below that of the average neighborhood in Kinshasa, which with few exceptions is impoverished. If there are “unique needs,” they most likely relate to the lack of services in this population prior to 2016 (for family planning and other social services), which resulted in the lower level of contraceptive use found in 2016. When asked this question directly, the co-author who is a high-level military official (HNE) replied that there are no special norms per se, but that this population is characterized by a high level of mobility; in particular, the unrest in the Eastern part of the country results in transfers of military personnel to those areas.

In the Methods, we have further described the setting in the military camps. In the Discussion, we have stated that “Findings from these two quantitative studies underscore the need for additional research using qualitative methods to better understand the dynamics on contraceptive use in this population.”

MEASURES: Other than the few instances where questions were only asked of military populations, or were only asked at one time point in the non-military survey, were the items identical? Did they use the same terminology and response options? If so, this should be explicitly stated. If not, this could be considered a limitation in trying to directly compare the two samples.

Reply: As noted above, we have added the following text to the Methods section:

The content and procedures for the PMA and military surveys were highly similar on the following points. The local study directors were the same for both surveys. The vast majority of the questions in the military surveys were taken verbatim from the PMA questionnaire, with the same terminology and response options, except for a few additional questions not found in the PMA questionnaire. The questions were administered by trained female interviewers in French or Lingala (the local language). Data were collected in both rounds and both populations using tablets or smartphones programmed with ODK. 

ANALYSIS: Perhaps it is because of the misplaced parenthesis on page 10, line 196, but the last few sentences of the analysis section are dense and difficult to read.

Reply: we have corrected this sentence to read:

Logistic regression generalized linear latent and mixed models (GLLAMM) with the logit link and binomial family[10] that adjusted for clustering and sampling weights were used to measure the level of association between outcomes and explanatory variables (association between contraceptive use and intended births while controlling for sociodemographic characteristics, reproductive history, and exposure to FP programming and messaging).

TABLES: Would help the reader to be clear that numbers in tables 1-3 are percentages unless specified as a mean (e.g., age). Also, it is not entirely clear why some significant results are bolded and others are not.

Reply: We have made this correction in the tables and have bolded all p values <0.05.

DISCUSSION: Some interesting findings I was hoping that discussion might address:

Reply: we have incorporated answers to most of these questions in different sections of the Discussion, as follows:

1) why were fewer military women living with husband in 2020 vs. 2016? 

“The composition of the women respondents in the military population differed on several variables between 2016 and 2019 (e.g., fewer women living with their husbands), whereas the characteristics of the non-military population showed less change. One reason may be the frequent deployment of military families to different camps across the country. Also, the sample size was far larger for the non-military respondents, contributing to less variability across years.”

2) Why were women across the board less knowledgeable about female condoms? 

“The lower knowledge of female condoms than other methods reflects more limited availability of this method in many service delivery sites.”

3) What happened to knowledge of rhythm method in military women? 

Note: we believe that the knowledge of the rhythm method among military women being much lower in 2019 than 2016 may have resulted from changes in the population over those three years (cited above). However, since we don’t cover knowledge of methods in the Discussion, we didn’t find a convenient place to insert this sentence and thus omitted it.

4) I noticed that the exposure sources included TV, radio, and magazine/newspaper. How is internet penetration within Kinshasa? Are these sources outdated? Are FP campaigns conducted by internet? Could less use of “outdated” sources account for drop in exposure in non-military?

Reply: Cellphone use has become almost ubiquitous in Kinshasa, but access to the internet is still limited to those in higher income levels. Although family planning messaging targeting adolescents and youth increasingly use social media, it has not been a major channel for reaching married women in the general population. Given the lower economic level among military families, it is even less likely that they would have access to the internet. Because the volume of FP messaging varies over time (often dependent on external donor funding), so may the percentage of the population reporting to have seen such messaging. 

Given that this reply does not change the text (and we are cognizant of the increased number of words in this manuscript, based on our revisions), we have not added this text to the Discussion section. If the editor/reviewer feel strongly on this point, we are happy to add it.

5) Why are more births unintended in military populations?

Reply: we allude to this in the Discussion section with the following comment: 

“Consistent with the lower contraceptive use among women in union in the military population, they are also less likely to report that their last birth was intended (45.5%) as compared to their non-military counterparts (58.1%).”

It is noted on line 329 that deployment of men may influence contraceptive choice; could the authors elaborate on how?

Reply: while we believe this statement to be true, we have no evidence to support this statement; thus, we have removed it from the text and added it as a question for future research. 

The authors seem to note that younger unmarried women in the PMA surveys showed higher levels of modern contraceptive use. Is there a reference that can be added for these findings? 

Reply: we have added the reference to PMA conducted in 2019-2020 to support this statement.

I think the last point about feasibility is important and an understated finding of this study. Perhaps that point could be highlighted along with explicit future research directions.

Reply: the last two paragraphs of the Discussion now address these questions.

Reply: our submission conforms to these instructions.

---

## [Decision Letter · Decision Letter 1]

26 May 2021

PONE-D-21-04034R1

The gap in contraceptive knowledge and use between the military and non-military populations of Kinshasa, DRC, 2016-2019

PLOS ONE

Dear Dr. Schulze,

Thank you for submitting your manuscript to PLOS ONE. After careful consideration, we feel that it has merit but does not fully meet PLOS ONE’s publication criteria as it currently stands. Therefore, we invite you to submit a revised version of the manuscript that addresses the points raised during the review process.

The two reviewers from the previous version accepted to review the revised manuscript. Reviewer 2 has still some minor comments to be addressed.

The data link to the military survey, http://doi.org/10.5061/dryad.k0p2ngf85, does not work.

The main concern on the editor side is that the revision did not take into proper consideration the main limitation raised in the previous version: That the military populations of 2016 and 2020 are different populations as it can be grasped from table 1. This is understandable, since military population might be deployed to different settlements, maybe some of the barracks have a specific “meaning” attached (eg: for new recruits, …). The fact is that the populations are different and that more can be done in the analysis regarding this.

1. It does not invalidate the analysis, but there should be no suggestions that the military population in the two periods is the same. The title changed, which is in the right direction since we do not know if there is a closing gap or the “new” military population comes from a different background? More examples that remain:

l. 43: “Similarly, modern contraceptive use increased significantly in both 44 populations, but the military women continued to trail their non-military counterparts on MCPR 45 (24.2% versus 29.7%.)”

l. 47: “by 2019 the women in the military population had 48 caught up to where the non-military population was as of 2016.”

You can use more neutral language talking about higher levels of contraceptive use or increase. You should avoid language that suggest change in behaviour “the women in …”: They are (or might be) different women. These two examples come from the abstract. Please revise thoroughly the text.

2. Also, more can be done to address the changing nature of the military population. Any additional information regarding the specific characteristics of the military population would be helpful. We know that the vast majority of women are not serving. It would be helpful to know who is serving (husband, son, father, …), length of residence in Kinshasa, duration of service in the military, place of origin (item G2), ethnic background. All these might influence the particular SRH needs of the military population vis a vis the general population. We also see a drastic change in the proportion living with the partners, but that does not necessarily indicate a different population: it might happen that the partner is deployed to a different setting while the woman stays. Information on ongoing conflicts or anything that might sed light would be helpful. It is stated “The same units of residence 162 were used in both military surveys, although the residents could have changed”. You have information on length of stay in the camp (item G4) and in Kinshasha (item G3). Please provide details on this. Maybe a table aside from table 1 looking more in detail at the military population, comparing 2016, 2020, and 2020 separated according to length of stay (4 years of more, or less).

3. You can also use this as a covariate to see differences in cp in the military population according to length of stay in Kinshasha, or compare as three different groups those in 2016, those in 2020 that were in 2016 and the newly arrived. The interpretation in terms of success of the military RH campaign is very different. Eg: You might observe a sharp increase in CP of those stayers indicating success of the programs but low CP of new entrants, or the reverse. Those two cases have two very different policy implications. Since this is connected to the main contribution of the paper it needs to be explored.

Other minor comments:

“Kinshasa, the capital city, has shown the usual effects 59 of urbanization on family size.”: The reader does not need to know what the “usual effects” are. Please describe what you mean, and if it is “usual”, provide a reference.

We look forward to receiving your revised manuscript.

Kind regards,

José Antonio Ortega, Ph.D.

Academic Editor

PLOS ONE

Journal Requirements:

Reviewers' comments:

Reviewer's Responses to Questions

**Comments to the Author**

1. If the authors have adequately addressed your comments raised in a previous round of review and you feel that this manuscript is now acceptable for publication, you may indicate that here to bypass the “Comments to the Author” section, enter your conflict of interest statement in the “Confidential to Editor” section, and submit your "Accept" recommendation.

Reviewer #1: (No Response)

Reviewer #2: All comments have been addressed

2. Is the manuscript technically sound, and do the data support the conclusions?

Reviewer #1: Yes

Reviewer #2: Yes

3. Has the statistical analysis been performed appropriately and rigorously? 

Reviewer #1: Yes

Reviewer #2: Yes

4. Have the authors made all data underlying the findings in their manuscript fully available?

Reviewer #1: Yes

Reviewer #2: Yes

5. Is the manuscript presented in an intelligible fashion and written in standard English?

Reviewer #1: Yes

Reviewer #2: Yes

6. Review Comments to the Author

Reviewer #1: The revision has addressed the vast majority of the reviewers concerns. I have two comments/concerns:

1. I would like further consideration of the influence of a prior unplanned pregnancy on use of modern contraception. On line 367 of the mark-up the authors the results section you comment on the higher rate of the most recent pregnancy being unintended among the military population. Then you say "This finding is consistent with the lower levels of knowledge and use among women in the military population." In the discussion section (line 485 of the mark up) you argue that the women in the military population "started from behind" in knowledge and use and the increase in knowledge and use is evidence of this population catching as a result of the family planning programing. If intendedness of the most recent pregnancy is a marker for lower knowledge and use, and I agree with you that it probably is along with use of less effective methods of contraception, then wouldn't this population be the most likely to benefit from family planning programing as they are starting from such low levels of knowledge? Why did you exclude this variable from your analyses?

2. In the discussion section (Line 530 of the mark up version), you discuss two papers that you are currently writing. I do not think this belongs in your manuscript. I think you can describe these topics as interesting opportunities for future research, but I don't think it is OK to warn off other researchers by describing papers that have not been published.

Reviewer #2: The authors have provided thoughtful and detailed responses to my comments. I specifically appreciate the greater level of detail in the methods that improve the clarity and reproducibility of this work, as well as their responsiveness to comments related to data analysis and discussion points.

7. PLOS authors have the option to publish the peer review history of their article (what does this mean?). If published, this will include your full peer review and any attached files.

Reviewer #1: **Yes: **Timothy A Roberts, MD MPH

Reviewer #2: No

---

## [Author Response · Author response to Decision Letter 1]

2 Jul 2021

Authors’ replies to the editors/reviewers’ comments: ONE-D-21-04034

(previous title) Closing the Gap: Contraceptive dynamics in military and non-military populations of Kinshasa, DRC, 2016-2020

(new title) The Gap in Contraceptive Knowledge and Use between the Military and Non-military populations of Kinshasa, DRC, 2016-2019

Note: in this document, all comments from the editors/reviewers are shown in bold (non-italics). All responses from the authors are shown in non-bold italics. In addition, we would like to draw attention to the change in the data of the second survey in our revised manuscript. Based on the actual dates of data collection, the surveys in both the military and non-military populations are much closer to a three- than a four-year interval. (Because the data collection for the second survey in both cases extended into 2020, we had opted for simplicity to describe it as a comparison of 2016 and 2019. We now specify that there was a three-year interval and (again for simplicity) have labeled the second surveys as occurring in 2019, although the exact dates are given in the methods section.

On May 26, 2021, the editor/reviewer sent us a series of new comments to address. Below are those points, as well as our replies. 

The data link to the military survey, http://doi.org/10.5061/dryad.k0p2ngf85, does not work.

Reply: we have experienced difficulty in posting the data set to Dryad. Instead, we are submitting it directly to the journal as supplementary file S6. If this is not satisfactory, please let us know. 

The main concern on the editor side is that the revision did not take into proper consideration the main limitation raised in the previous version: That the military populations of 2016 and 2020 are different populations as it can be grasped from table 1. This is understandable, since military population might be deployed to different settlements, maybe some of the barracks have a specific “meaning” attached (eg: for new recruits, …). The fact is that the populations are different and that more can be done in the analysis regarding this. 

Reply: we concur and have edited the text consistent with the points below.

1. It does not invalidate the analysis, but there should be no suggestions that the military population in the two periods is the same. The title changed, which is in the right direction since we do not know if there is a closing gap or the “new” military population comes from a different background? More examples that remain:

l. 43: “Similarly, modern contraceptive use increased significantly in both 44 populations, but the military women continued to trail their non-military counterparts on MCPR 45 (24.2% versus 29.7%.)”

Reply: we have reworded the results section of the abstract to read: 

The socio-demographic profile of women in the military camps differed between 2016 and 2019, which may reflect the more mobile nature of this population. In both populations, knowledge of modern contraceptive methods increased significantly. Similarly, use of a modern contraceptive method also increased significantly in both, though by 2019 women in the military camps were less likely to use modern contraception (24.9%) than their non-military counterparts (29.7%). Multivariate analysis showed no significant difference in the amount of increase in MCPR for the two populations. Among contraceptive users in both populations, the implant was the leading method. Potential effects of FP programming were evident in the military population: exposure to FP messaging increased (in comparison to a decrease among the non-military population). Moreover, women who had lived in the camps for 4+ years had a higher MCPR than those living in the camps for less than four years. 

l. 47: “by 2019 the women in the military population had caught up to where the non-military population was as of 2016.”

Reply: see above; line removed.

You can use more neutral language talking about higher levels of contraceptive use or increase. You should avoid language that suggest change in behaviour “the women in …”: They are (or might be) different women. These two examples come from the abstract. Please revise thoroughly the text.

Reply: we have attempted to simply describe the military population in 2019 without implying a change in behavior. For example, when referring to the military population, in several places we have removed “increased significantly” and replaced it with “was significantly higher in 2019 than in 2016.” However, this change results in extra words that do not read as smoothly as “increased.” If the editor or reviewer feels that we need to remove ALL MENTION of “increased” among the military population, we can do so. However, we would argue that the additional language around the differences between surveys in the military population should be sufficient to communicate the desired message. 

2. Also, more can be done to address the changing nature of the military population. Any additional information regarding the specific characteristics of the military population would be helpful. We know that the vast majority of women are not serving. It would be helpful to know who is serving (husband, son, father, …), length of residence in Kinshasa, duration of service in the military, place of origin (item G2), ethnic background. All these might influence the particular SRH needs of the military population vis a vis the general population. 

Reply: we have added data in Table 1 and in the text on the following variables for the military population: length of residence in military camps, length of residence in Kinshasa, percent reporting to have been interviewed in 2016, percent born in Kinshasa. We did not have information on the other variables listed above (relationship to the person serving in the military, place of origin, ethnic background).

We also see a drastic change in the proportion living with the partners, but that does not necessarily indicate a different population: it might happen that the partner is deployed to a different setting while the woman stays. Information on ongoing conflicts or anything that might sed light would be helpful. It is stated “The same units of residence 162 were used in both military surveys, although the residents could have changed”. You have information on length of stay in the camp (item G4) and in Kinshasa (item G3). Please provide details on this. Maybe a table aside from table 1 looking more in detail at the military population, comparing 2016, 2020, and 2020 separated according to length of stay (4 years of more, or less).

Reply: we have added information to Table 1 and in the text on the length of residence in Kinshasa and in the military camps as well as percent born in Kinshasa.

3. You can also use this as a covariate to see differences in cp in the military population according to length of stay in Kinshasa, or compare as three different groups those in 2016, those in 2020 that were in 2016 and the newly arrived. The interpretation in terms of success of the military RH campaign is very different. Eg: You might observe a sharp increase in CP of those stayers indicating success of the programs but low CP of new entrants, or the reverse. Those two cases have two very different policy implications. Since this is connected to the main contribution of the paper it needs to be explored.

Reply: we have added the suggested comparison of the 3 groups: 2016, 2019-residence in military camps of 4+ years, and 2019-residence of <4 years; see Figure 2. 

Other minor comments:

“Kinshasa, the capital city, has shown the usual effects of urbanization on family size.”: The reader does not need to know what the “usual effects” are. Please describe what you mean, and if it is “usual”, provide a reference.

Reply: we have added two citations.

White MJ, Muhidin S, Andrzejewski C, Tagoe E, Knight R, Reed H. Urbanization and fertility: an event-history analysis of coastal Ghana. Demography. 2008;45(4):803-16. Epub 2008/12/30. doi: 10.1353/dem.0.0035. PubMed PMID: 19110898; PubMed Central PMCID: PMCPMC2834382.

Lerch M. Regional variations in the rural-urban fertility gradient in the global South. PLoS One. 2019;14(7):e0219624. Epub 2019/07/20. doi: 10.1371/journal.pone.0219624. PubMed PMID: 31323039; PubMed Central PMCID: PMCPMC6641161.

I would like further consideration of the influence of a prior unplanned pregnancy on use of modern contraception. On line 367 of the mark-up the authors the results section you comment on the higher rate of the most recent pregnancy being unintended among the military population. Then you say "This finding is consistent with the lower levels of knowledge and use among women in the military population." In the discussion section (line 485 of the mark up) you argue that the women in the military population "started from behind" in knowledge and use and the increase in knowledge and use is evidence of this population catching as a result of the family planning programing. If intendedness of the most recent pregnancy is a marker for lower knowledge and use, and I agree with you that it probably is along with use of less effective methods of contraception, then wouldn't this population be the most likely to benefit from family planning programing as they are starting from such low levels of knowledge? Why did you exclude this variable from your analyses?

Reply: we introduced the variable “unintended pregnancy” into the analysis of factors affecting MCPR. It was not significant, and it reduced the power because it reduced the number of cases from 2517 to 1873 (omitted were women who had not had a pregnancy in the last five years or were not currently pregnant). We could mention this finding in the text, but our strong preference is to simply omit discussion of unintended pregnancy in this context. 

2. In the discussion section (Line 530 of the mark up version), you discuss two papers that you are currently writing. I do not think this belongs in your manuscript. I think you can describe these topics as interesting opportunities for future research, but I don't think it is OK to warn off other researchers by describing papers that have not been published.

Reply: We have removed the two sentences in question about the other two articles.

---

## [Editor Report · Decision Letter 2]

7 Jul 2021

The gap in contraceptive knowledge and use between the military and non-military populations of Kinshasa, DRC, 2016-2019

PONE-D-21-04034R2

Dear Dr. Schulze,

We’re pleased to inform you that your manuscript has been judged scientifically suitable for publication and will be formally accepted for publication once it meets all outstanding technical requirements.

Kind regards,

José Antonio Ortega, Ph.D.

Academic Editor

PLOS ONE

Additional Editor Comments (optional):

It is felt that it is not necessary to send the manuscript back to the reviewers. The changes have improved the manuscript.

There is one change that has not been incorporated and must be incorporated before publication.

In the previous revision it was commended:

"“Kinshasa, the capital city, has shown the usual effects 59 of urbanization on family size.”: The reader does not need to know what the “usual effects” are. Please describe what you mean, and if it is “usual”, provide a reference."

You have provided references. But the inconcrete statement is still there. Please specify what you mean by usual effects. Then you can qualify, e.g: "as observed in other settings" and provide the reference.
---

## [Editor Report · Acceptance letter]

19 Jul 2021

PONE-D-21-04034R2 

The Gap in Contraceptive Knowledge and Use between the Military and Non-military populations of Kinshasa, DRC, 2016-2019 

Dear Dr. LaNasa:

I'm pleased to inform you that your manuscript has been deemed suitable for publication in PLOS ONE. Congratulations! Your manuscript is now with our production department. 

Kind regards, 

on behalf of

Dr. José Antonio Ortega 

Academic Editor

PLOS ONE